# Operative Protocol for Testing the Efficacy of Nasal Filters in Preventing Airborne Transmission of SARS-CoV-2

**DOI:** 10.3390/ijerph192113790

**Published:** 2022-10-23

**Authors:** Sabrina Semeraro, Anastasia Serena Gaetano, Luisa Zupin, Carlo Poloni, Elvio Merlach, Enrico Greco, Sabina Licen, Francesco Fontana, Silvana Leo, Alessandro Miani, Francesco Broccolo, Pierluigi Barbieri

**Affiliations:** 1INSTM National Interuniversity Consortium of Materials Science and Technology, Research Unit of University of Trieste, 34127 Trieste, Italy; 2Department of Chemical and Pharmaceutical Sciences, University of Trieste, Via L. Giorgieri 1, 34127 Trieste, Italy; 3Institute for Maternal and Child Health, IRCCS Burlo Garofolo, Via dell’Istria 65/1, 34137 Trieste, Italy; 4Department of Engineering and Architecture, University of Trieste, Via A. Valerio 10, 34127 Trieste, Italy; 5SIMA Società Italiana di Medicina Ambientale, Viale di Porta Vercellina, 9, 20123 Milano, Italy; 6Ospedale San Polo, Azienda Sanitaria Universitaria Giuliano Isontina, Via Luigi Galvani 1, 34074 Monfalcone, Italy; 7Division of Oncology, Vito Fazzi Hospital, P.za Muratore 1, 73100 Lecce, Italy; 8Department of Environmental Science and Policy, University of Milan, Via Festa del Perdono 7, 20122 Milano, Italy; 9Department of Medicine and Surgery, School of Medicine, University of Milano-Bicocca, 20900 Monza, Italy; 10Cerba HealthCare Italia, Via Durini, 14, 20122 Milano, Italy

**Keywords:** SARS-CoV-2 airborne transmission, endonasal filters, viral filtration efficacy protocol, bio-gel AgNP filters

## Abstract

Background: Standardized methods for testing Viral Filtration Efficiency (VFE) of tissues and devices are lacking and few studies are available on aerosolizing, sampling and assessing infectivity of SARS-CoV-2 in controlled laboratory settings. NanoAg-coated endonasal filters appear a promising aid for lowering viable virus inhalation in both adult and younger populations (e.g., adolescents). Objective: to provide an adequate method for testing SARS-CoV-2 bioaerosol VFE of bio-gel Ag nanoparticles endonasal filters, by a model system, assessing residual infectivity as cytopathic effect and viral proliferation on in vitro cell cultures. Methods: A SARS-CoV-2 aerosol transmission chamber fed by a BLAM aerosol generator produces challenges (from very high viral loads (10^5^ PFU/mL) to lower ones) for endonasal filters positioned in a Y shape sampling port connected to a Biosampler. An aerosol generator, chamber and sampler are contained in a class II cabinet in a BSL3 facility. Residual infectivity is assessed from aliquots of liquid collecting bioaerosol, sampled without and with endonasal filters. Cytopathic effect as plaque formation and viral proliferation assessed by qRT-PCR on Vero E6 cells are determined up to 7 days post inoculum. Results: Each experimental setting is replicated three times and basic statistics are calculated. Efficiency of aerosolization is determined as difference between viral load in the nebulizer and in the Biosampler at the first day of experiment. Efficiency of virus filtration is calculated as RNA viral load ratio in collected bioaerosol with and without endonasal filters at the day of the experiment. Presence of infectious virus is assessed by plaque forming unit assay and RNA viral load variations. Conclusions: A procedure and apparatus for assessing SARS-CoV-2 VFE for endonasal filters is proposed. The apparatus can be implemented for more sophisticated studies on contaminated aerosols.

## 1. Introduction

SARS-CoV-2 (Severe Acute Respiratory Syndrome Coronavirus type 2) is the etiological agent of the respiratory COronaVIrus Disease 2019 (COVID-19) pandemic that from the beginning of the 2020 has spread worldwide, evolving into several variants of concern (VOCs) that have troubled the human population and the public health sector. The current more frequent variant is the Omicron (B.1.1.529), with BA.1, BA.2, BA.3, BA.4 and BA.5 as the designated descendant variants. Despite the higher transmissibility of Omicron SARS-CoV-2 BA-1 and BA.2 with respect to the previous variants, a decrease in disease severity is detected in the individuals infected [1]. Actually, BA.2, BA.4 and BA.5 are classified as variants of concern (VOCs) by the European Centre for Disease Prevention and Control (ECDC), BA.5 being the predominant sub-strain circulating in the UE; nevertheless, ECDC does not indicate an effect on severity and transmissibility, although the number of cases is continuously increasing [2]. Nevertheless, a negative impact on vaccine efficacy is regrettably denoted for these SARS-CoV-2 Omicron sub variants [3]. Moreover, a variant has evolved from variant under monitoring (VUM) to variant of interest (VOI), i.e., omicron BA.2.75. That being said, it arises that SARS-CoV-2 develops quite rapidly and the control of future strains should be mandatory to contain the pandemic, since it cannot be excluded that new variants might be more transmissible and produce more severe disease. Despite the increased transmissibility, in vitro studies have shown that the Omicron strain seems to replicate more slowly than Delta [4] and induces less syncytia formation [5]. Interestingly, Omicron spike protein (structural study [6]) and Omicron virions (functional study [7]) are more stable with respect to previous strains. It has also been estimated in a computer modeling study [8] that Omicron variant airborne transmission contributed to 34–38% of asymptomatic-pre-symptomatic phase diagnose, with respect to a negligible contribution (less than 10%) to the symptomatic COVID-19 phase, when droplet transmission is predominant (asymptomatic-pre-symptomatic stage: 21–28%; symptomatic stage: 48–71%); finally, the contact role is similar in the two disease conditions (asymptomatic-pre-symptomatic stage: 37–45%; symptomatic stage: 25–42%). This difference could be due to the size of particle emitted; indeed, in asymptomatic-pre-symptomatic infection, the virus was spread through breathing and talking (droplets < 10 μm), while during the symptomatic disease, coughing and sneezing predominates, characterized by larger dimensions that tend to deposit on surfaces. Studies of both indoor and outdoor air viral contamination require definition of protocols regarding sampling criteria [9], detailing instruments, duration, positioning and eventual sample storage [10].

Despite SARS-CoV-2 being a respiratory virus, studies on its aerosol transmission in controlled laboratory settings are still limited and standardized national or international viral filtration efficiency testing protocols are still missing [11,12]. Studies have mainly used bacteriophage and influenza viruses or inactivated viruses as challenges in filtration tests [11,12,13,14,15]. Actually, the indications to prevent transmission among the population include social distancing, the avoiding of people-gathering situations, ventilation of the indoor environment and personal and environmental hygienic measures [16]. Not in all real-world situations can these measures be easily afforded; as an example, schools are indoor environments at high risk due to the long-term gathering of children and adolescents [17]. ECDC recommended the promotion of physical distancing through cohorting of groups and classes (on the base of infection risk and health status), distance within the classroom (e.g., spacing desks), reduction of class size, staggering the start and the end of the lessons and break times, or holding lessons in outdoor environment. Nevertheless, some of these indications are not feasible in all schools and situations [18].

Last but not least, Personal Protective Equipment (PPE), such as medical masks and respirators, have been and still represent a key feature in limiting the spread of SARS-CoV-2.

With the employment of PPE, it is possible to reduce the penetration of pathogens through the reduction of the “microbial load” in inhaled air, so decreasing the risk of transmission of many respiratory infectious agents in occupational or community settings. Moreover, the constituents of indoor air that may be involved in the exacerbation or in the induction of respiratory infections can be trapped. Filtering face respirators, hygienic and surgical masks designed as PPE, can decrease the inhaled microbial load through a simple filtration process. On the other hand, these devices trap microorganisms at the expense of a reduction in the inhaled airflow, so impacting negatively on the respiratory capacity [19]. Compared to currently available PPEs, endonasal filters [20] with silver nanoparticles (AgNPs) possess some advantages, due to the combination of the antibacterial and antiviral action of AgNPs [21] with common filtration processes. This dual mechanism reduces microbial infectivity and is able to protect the lower airways. They are expected to be used when nose airways are clean and without symptoms of rhinitis. Biogel-AgNP nasal filters can trap and inactivate bacteria and viruses, so they can be considered a promising type of PPE to be employed both in occupational and community settings [22]. In the healthcare environment, bio-gel-AgNP nasal filters may be used to prevent the transmission of microbial agents between patients and healthcare personnel, but their use can be extended to other occupational settings such as schools, universities, offices, or where a possible contamination by biological agents and/or risk of microbial transmission to humans exist. Furthermore, in the general population (both adult and younger, e.g., adolescents), bio-gel-AgNP nasal filters can contribute to the inhibition of microbial agents# transmission through air in domestic and outdoor environments, as well as when masks are not worn, e.g., break times and lunchtime at school, workplace canteens. Biogel-AgNP nasal filters can thus contribute to the prevention of air-transmitted infectious diseases, guaranteeing long time continuous use combined with breathing comfort, for day long protection in both adult and younger populations. In order to test the efficacy of bio-gel-AgNP nasal filters in blocking SARS-CoV-2 transmission, in the following an experimental protocol for assessing SARS-CoV-2 infectivity reduction by endonasal filters will be described, where SARS-CoV-2 is aerosolized in a controlled setting inside a class II cabinet hood in a Biosafety level 3 facility.

## 2. Materials and Methods

### 2.1. Rationale and Specific Aims

The present protocol evolves from our experimental model of SARS-CoV-2 aerosol generation and transmission employing a BLAM bioaerosol generator, a cylindrical chamber for aerosol travel and a swirling bioaerosol collector SKC Biosampler for the collection of the particles [23], and from the experimental model for assessing the SARS-CoV-2 filtering efficacy of face masks published during 2020 by Ueki and coworkers [24]. An 8-Jet BLAM nebulizer operated as Multi Pass Atomizer mimics the spread of virus particles from a COVID-19 positive emitter patient (8 L per minute) while the SKC Biosampler sucks air at flow comparable to that of human breath inspiration (12.5 L per minute). SARS-CoV-2 is aerosolized into a plexiglass parallelepiped box (100 × 40 × 50 cm) positioned in a class II cabinet hood in a Biosafety level 3 laboratory and collected through a Y shaped tube sampling port—simulating nostrils—connected to the SKC sampler. Such a sampling port is a simple model for testing the mechanical filtration performance of AgNP coated endonasal filters, as well as reduction of microbial infectivity by flow interaction with internal coated filter surfaces. The sampling setting can be modified for testing facial masks by inserting a human-like head within the bioaerosol chamber, as in [24]. Infective virus concentration is determined by use of a plaque assay and the RNA viral load quantified by employing quantitative real-time reverse transcription PCR (qRT-PCR). The efficiency of the bio-aerosol nebulization and transmission system is assessed after the aerosolization determining the viral load in the SKC sampler compared to that remaining in the nebulizer. Effectiveness of Biogel silver nanoparticle (AgNP) endonasal filters inserted into the Y shaped tube sampling port, connected to the SKC sampler simulating nose inhalation, is evaluated, measuring infective viruses that pass through the endonasal filters and are collected in the SKC bio-sampler reservoir, by use of a plaque forming unit (PFU) assay and quantitative reverse transcription real-time PCR (qRT-PCR) [23].

### 2.2. Experimental Setting and Procedure

A model for SARS-CoV-2 air contamination, airborne transmission and viral filtration efficiency testing is constituted by a plexiglass test chamber (100-cm long × 40-cm wide × 50-cm high—Figure 1) with removable circular plexiglass closures, holed to allow bioaerosol input (from BLAM bioaerosol generator) and collection (by SKC Biosampler) in the opposite small sides of the test chamber. This device represents the first containment for viral bio-aerosol.

The test chamber is positioned within the Class II biosafety hood in a BSL3 laboratory, together with BLAM generator and SKC Biosampler with backup NaClO impinger and drier for protecting the sampling pump (as described in detail in [23]). This setting represents the second higher level of viral containment, positioned in a BSL3 laboratory (third higher level of containment).

The SKC Biosampler collects the bioaerosol from the chamber through a 3D-printed Y shaped PTFE tube sampling port simulating nostrils (small internal diameter 10 mm; large size i.d. 15 mm; length 80 mm) that can host the endonasal filters to be tested. NOTE: in advanced experiments, it can be connected to the SKC Biosampler by different length tubes (inert glass or PTFE) modifying distance between bioaerosol generation port and Y sampling port, allowing simulation of different distances between the infected emitter and the susceptible receiver in a poorly dispersive environment. Distances can be set, e.g., as 25 cm, 50 cm, 95 cm.

Proper directional airflow through the safety cabinet containing the test chamber, and the sealability of the test cabinet can be checked by using smoke tests. The relative humidity (RH) and temperature in the test chamber within the safety hood are at 75 ± 15% and 20 ± 1 °C, respectively. The BLAM nebulizer is charged with 6 mL of virus suspension to generate droplets/aerosols, emitted continuously for 15 min, simulating transmission from an infected individual. This time point is chosen, as the ECDC, CDC and WHO guidelines indicated that 15 min are the cut-off to define the close contact between COVID-19 positive individuals and susceptible persons in a closed environment, without PPE [25,26,27]. Virus suspensions can be (1) 10^5^ PFU/mL (~10^7^ RNA viral copies/mL) to simulating very high viral load [23,28,29] and/or (2) 10^4^ PFU/mL (~10^6^ RNA viral copies/mL), considered by [26] in simulations for risk assessment. The number of particles emitted by the 8-Jet BLAM nebulizer operated as Multi Pass Atomizer is approximately one million and is compatible with the number of particles generated by multiple coughs or by a single sneezing. The size distribution from the aerosol generator is described in [21] with modes at 0.5 and 2 μm. The SKC Biosampler sampling flow (12.5 L per minute) is comparable to inhalation during moderate activity and it occurs through the 3D-printed Y shaped PTFE tube sampling port (Figure 2). The section of PTFE tube can be 15 mm or 10 mm, simulating wider or narrower nostrils that host different sizes of endonasal filters (Figure 3).

A first set of experiments (without the filters) is necessary to test the system’s efficiency in virus aerosolization and collection and would simulate a situation of close contact without PPE. The more diluted virus suspension is tested first to reduce the probability of carry-over. Three replicate experiments are performed for each of the two virus suspensions. BLAM nebulizes virus for 15 min and contemporaneously the SKC bio-sampler collects the aerosol from the chamber through the 3D-printed Y shaped PTFE tube sampling port. One of the removable side circular bioaerosol chamber closures hosts a hole with a 0.2 um filter to allow balance in the aerosol chamber between inlet (aerosol generator, 8 lpm) and outlet (sampler, 12.5 lpm) flows.

Between the experimental replicates, the BLAM and the SKC bio-sampler reservoirs are carefully washed with sterile water; moreover, the chamber and the collecting tubes are flushed for 5′ with 75% ethanol and 5′ with sterile water to avoid carryover between the replicates. A further 10’ of air flushing is added in order to remove ethanol residuals from the bioaerosol chamber. At the end of the replica tests, the chamber and the collecting tubes are flushed for 15′ with 75% ethanol and 15′ with sterile water and the chamber is cleaned internally with ethanol and water. Prior to each experimental replica, the presence of viral RNA in the SKC Biosampler will be tested in order to verify the correct disinfection of the apparatus.

A second set of experiments is needed to test the effectiveness of the endonasal filter in preventing the droplet/aerosol transmission of virus. A couple of endonasal filters (Small/Large size) are inserted into the two branches of the Y shape connector of 10–15 mm internal diameter to prevent the droplet/aerosol transmission of viable virus to the Biosampler collection liquid. BLAM nebulizes virus for 15 min and contemporaneously the SKC bio-sampler collects the aerosol from the chamber through the 3D-printed Y shaped PTFE tube sampling port. Three replicate experiments are performed with endonasal filters on the sampling line for each of the two virus suspensions. To prevent cross-contamination of the infectious virus and viral RNA, the chamber and the collecting tubes are flushed for 5′ with 75% ethanol and 5′ with sterile water and all filters are disposed after each experimental trial. Virus aerosolization and quantification follow [21].

#### 2.2.1. SARS-CoV-2 Suspension Preparation

The SARS-CoV-2 selected variant isolated, amplified and quantified on Vero E6 cell line, is employed in the aerosolization [30,31,32]. The virus is diluted at 10^5^ and 10^4^ PFU/mL for aerosol generation in the infection medium composed by MEM supplemented with 2 mM glutamine, 2% fetal bovine serum and 100 U/mL penicillin/streptomycin.

#### 2.2.2. The Bio-Aerosol Measuring Train

Experiments on residual infectivity after virus aerosolization are conducted in a class II biosafety cabinet within a BLS3 facility, in a sealed experimental bio-aerosol chamber assembled as described above. A BLAM aerosol generator hosting the SARS-CoV-2 suspension in its precious jar received an air flow produced by an AERO Particle Nebulizer pump (TCR Tecora Srl—Cogliate, MB, Italy) positioned outside the cabinet which generate the bio-aerosol. The BLAM has a filtered inlet of 0.2 μm, allowing entrance of air needed to sustain the air flow in the measuring train. The aerosol is transferred into the bioaerosol chamber described above, and then to a Biosampler (SKC Inc., Eighty Four, PA, USA), which collect the aerosolized SARS-CoV-2 into a dedicated vessel. An aspiration flow of 12.5 L per minute generated by a Bio-Bravo sampling pump (TCR Tecora Srl, Cogliate, MB, Italy) sustains the virus particle collection into the Biosampler. A glass impinger, with sodium hypochlorite working as safety trap deactivating eventual unsampled pathogens, and a silica dryer are positioned online between the Biosampler and the Bio-Bravo to protect the pump from vapors.

All the connections are prepared in nylon with o-rings. The bio-aerosol measuring train (Figure 1) has been set up and tested with *E. coli* BL21-DE3.

#### 2.2.3. Aerosol Generation

The selected bioaerosol generator is a Blaustein Atomizing Modules (BLAM) nebulizer (CH Technologies Inc., Westwood, NJ, USA) with 8 jets, in consideration of superior nebulization efficiency [33] in comparison to common Collision nebulizers or the Sparging Liquid Aerosol Generator (SLAG). The BLAM generates less significant impacts of the bioaerosol on hard surfaces and requires lower pressures than the Collision nebulizer, provoking less stress to the microorganisms, with a resulting inferior loss of infectivity. The virus suspension (~5 mL) is added in the precious jar of an 8-jet BLAM nebulizer with horizontal discharge, operated in Multi Pass Atomization mode, at a flow of 8 L per minute (lpm). The aerosol is released in a 1000 × 500 × 400 mm parallel-piped bioaerosol chamber, subject to aspiration from the Biosampler (12.5 lpm); the duration of aerosolization is 15 min. Filtered air required for equilibrating flows enters into the system through a 0.2-micron Poly-ethersulfone filter fitted onto the BLAM cap and on a further hole on the removable side closure of the parallel pipes. After aerosolization, the remaining liquid is collected in a falcon tube and the chamber is washed with water; after that, the viral suspension for the next run is added.

#### 2.2.4. Size Distribution Temperature and Relative Humidity Assessment

The size distribution of aerosol particles generated from the infection medium (MEM + 2% fetal bovine serum; 2 mM glutamine; 100 U/mL penicillin/streptomycin) by the BLAM nebulizer is measured in dedicated runs at the inlet and outlet of the parallel-piped aerosol chamber, with 8 lpm aspiration from TCR Tecora Bio Bravo, to check size distribution and eventual modifications. An Optical Particle Counter measures size distribution (e.g., GRIMM EDM) counting 10^7^ particles in the range 0.25–32 μm, with acquisition time of 6 s. Relative humidity (RH) and temperature are measured using a thermo-hygrometer (e.g., Bluetooth connected Inkbird IBS- TH1 Plus), located in the chamber.

#### 2.2.5. Bioaerosol Sampling

For the collection of viable viruses from aerosol, a swirling aerosol collector—e.g., BioSampler (SKC Inc., Eighty Four, PA, USA)—connected by the Y shaped tubing to the end of the parallel-piped aerosol transmission chamber is used for sampling bioaerosol in 20 mL of infection medium. The sampling duration is 5 or 10 min. The sampling flow is 12.5 lpm sustained by a TCR Tecora Bio Bravo pump. The pump is protected from microbial contamination by an impinger containing NaClO as a disinfectant and by a silica dryer that treats the sucked air. At the end of the procedure, the liquid sample is collected in a falcon tube; the sample container is washed in distilled water and refilled with a fresh medium.

#### 2.2.6. Infectivity Assessment

The infectivity of the bio-aerosol samples is determined in vitro on Vero E cells (epithelial kidney standard cell line derived from *Cercopithecus aethiops*). Vero E6 cells still represent a gold standard for viral infection experiments and are highly susceptible to SARS-CoV-2 infection [31].

Vero E6 cells are maintained in MEM supplemented with 2 mM glutamine, 10% fetal bovine serum and 100 U/mL penicillin/streptomycin and seeded on 6 multi-well plates (350.000 cells for well). During the infection phase, the cells will be cultured in 2% FBS. Briefly, after the sampling, the liquid collected at the end of the experimental setting is filtered with a 0.2-μm filter in order to eliminate bacteria and impurities potentially present in the apparatus.

The samples are seeded on a monolayer of Vero E cells and monitored by the optical microscope for 6 days to assess morphological modifications, e.g., cytopathic effects (CPE) characterized by cell vacuolization, rounding and detachment.

The viral load in the supernatant is determined through semi-quantitative real time PCR (qRT-PCR) daily. The RNA is extracted through thermolysis; 15 μL of the supernatants is mixed with 45 μL of distilled water and heated at 98 °C for 3′, followed by 5′ at 4 °C. The samples are then stored at −80 °C to avoid RNA degradation until analysis.

The intracellular RNA is extracted with an RNA Zymo RNA extraction kit (Zymo Research, Irvine, CA, USA). qRT-PCR is performed with The Luna Universal Probe One-Step RT-qPCR Kit (New England Biolabs, Ipswich, MA, USA) with the primers and probes targeting gene N, E and sub-genomic E (the last to detect the viral replication intermediates indicative of active viral amplification inside the cells [34,35]) as reported in Table 1.

A plaque forming unit assay is also performed.

100 μL of the collected sample is seeded in triplicate on a monolayer of Vero E6 cells, then after 1 h the cells are overlaid by immobilizing medium composed by 1:1 DMEM + 4% FBS: carboxy methyl cellulose (CMC) 4% and incubated until the plaques are visible (4–6 days). At the end of the 4–6 days, 1 mL of PBS is added in each well to remove the immobilizing medium. After washing, the cells are fixed with paraformaldehyde 4% in PBS for 20 min and stained with 0.1% crystal violet in PBS (Merck KGaA, Darmstadt, Germany) for 1 h. After 3 washes in water, the plates are air-dried, and the plaques formed are counted.

#### 2.2.7. Viral RNA Load after Aerosolization

SARS-CoV-2 RNA viral load is measured in the liquid present in the initial suspension, in the nebulizer after aerosolization and in the aerosol samples collected in the SKC bio-sampler, by the procedure described in the previous chapter.

### 2.3. Statistical Analysis

Each experimental setting (viral load; size of endonasal filter) is replicated three times and for each replica set, median (average) and standard deviation are calculated.

The efficiency of aerosolization is determined by evaluating the difference between the RNA viral load in the nebulizer (initial viral concentration) and in the SKC bio-sampler at day 0 (day of experiment).

The efficacy of virus filtration is calculated as the RNA viral load ratio between the collection of viruses with and without filters applied to the port of the collection tube of the SKC bio-sampler at day 0 (day of experiment). Other time points are also tested.

The growth curves, measured daily through molecular tests, are compared between the virus collected with and without filters through linear regression analysis.

The intracellular viral load at day 7 (end of the experimental planning) and the results from PFU assay between the virus collected with and without filters is compared by Mann-Whitney test.

## 3. Evaluation Outcomes

The principal aim of the study is the setup of safe apparatus and procedure for the assessment of the filtration power of bio-gel silver nanoparticle (AgNP) endonasal filters on SARS-CoV-2 viruses.

Evaluations at different stages of the procedure are mandatory to define the performance of our system and represent the checking points for the quality assessment of the study protocol.

The efficiency of the aerosolization and collection systems will be assessed on the first day of experiment, as well as the efficacy of virus filtration by endonasal filters. The assessment is determined by the measurement of RNA viral load between the reservoir of the nebulizer and the jar of the air-sample and between the samples collected by air-sampler with and without the endonasal filter.

The viral load abatement resulting from endonasal filter application would be shown as the percentage of the RNA level reduction.

Other time points are studied in order to determine if the endonasal filters block infectious virions. Indeed, the molecular determination of viral RNA does not underline the infectious potential of the sample collected.

To achieve this task the samples collected in the SKC bio-sampler are seeded on Vero E6 cells and monitored daily. The presence of cytopathic effect, the formation of plaque in the PFU assay and the increment of the viral load in the supernatant are all indicators of virus amplification that will be taken into consideration in the study. The presences of viral genomic RNA and of sub-genomic RNA intracellularly are also assayed. The sub-genomic RNA is an intermediate of intracellularly viral replication machinery and a useful marker to check the virus replication.

## 4. Discussion

The protocol here presented would suggest some basic useful information to set up a method for testing PPE in a dynamic status (i.e., the nebulization of the virus and aerosol sampling) mimicking as much as possible a real-life situation.

SARS-CoV-2 spreading has been exhausting health system capacity. Although the predominant Omicron variants have been leading to more attenuated symptoms and to the reduction of the hospitalization rate, the ability of Omicron SARS-CoV-2 to evade the immune system greatly impacts on the continuously increased number of infected individuals [1,2,3].

Therefore, the need for testing PPE to counteract virus spread continues to be a challenging topic. Moreover, there is no standard procedure for the quality control of PPE, employing infectious virus and dynamic conditions.

The main result that this study will achieve is the setting up of a standardized method to test PPE in laboratory-controlled conditions.

Our study protocol will accurately describe all the steps of the procedure to analyze PPE (in this case bio-gel silver nanoparticle (AgNP) endonasal filters) efficacy in blocking the virus passage.

The procedure here characterized will strictly simulate the aerosol emission from an infected person (BLAM nebulizer with a flow of 8 L per minute) in a close environment (the aerosol chamber) for the period of close contact, definition (15 min), to a susceptible host (the PTFE tube sampling port, 10 or 15 mm simulating wider or narrower nostrils, connected to the SKC bio-sampler with a flow of 12.5 L per minute).

By employing this setting, the efficiency of the system in generating and collecting bio-aerosol can be defined as well as the performance of virion retainment by the bio-gel silver nanoparticle (AgNP) endonasal filters.

We expect that this study protocol can be easily translated into further PPE testing applications and will contribute to defining standard conditions for dynamic quality control. Indeed, the experimental setting may be modified for assessing filters or masks optimized for children (pediatric applications) and teenagers, by modulating the aspiration of the samplers (less than 12.5 lpm) and using filter holders with a lower section; such variations can modify inner flows in the filters and the effectiveness of interaction of bioaerosol/airborne pathogens with the AgNP coated walls. It will also be possible to simulate the inhalation of children, adolescents, adults or the elderly, with screening conditions typical of different ages.

The main limit of the study is related to the controlled laboratory condition tested. Our setting is a simulation of a real-life situation and the absolute efficacy of endonasal filters can only be tested live. Nevertheless, our system can be very useful in obtaining indications on the efficacy of endonasal filters in blocking virus passage, at least in vitro. Moreover, all the steps (virus concentration, virus aerosolization, aerosol sampling, duration of the test, flow speed) are carefully selected in order to be as similar as possible to a critical real indoor environment. Another limit is related to the traslatability of the results in a real setting, in terms of virion numbers that are really able to infect the host and so produce illness.

Up to now, no standard test method is available for assessing viral filtration or inactivation efficiency of PPE for airborne SARS-CoV-2 [13]. An adapted protocol testing viral filtration and inactivation efficiency of face masks has been proposed by Nelson Laboratories starting from a Bacterial Filtration Efficiency test ASTM F2101-14 and using PhiX174 virus as a challenge [36], with mean particle size of 3.0 ± 0.3 µm and collection of the bioaerosol on a six-stage Andersen impactor; PhiX174 is also an enveloped virus considered a possible nonpathogenic surrogate for SARS-CoV-2 [37], and MS2 bacteriophages have been proposed [15,38], or inactivated virus [17].

Viral residual infectivity transmitted through masks or filters can be more properly assessed by aerosolization of viruses of interest, without resorting to surrogates. Like our protocol, a work by Ueki et al. [24] illustrated an airborne transmission simulator, where the air was sampled by using gelatin membrane filters and different types of masks were tested. Interestingly, SARS-CoV-2 was able to pass through cotton and surgical masks but also through N95 masks and fitted N95 (completely fitted with adhesive tape to the mannequin head), although the researchers showed a barrier effect for all the types of PPE, higher when the mask was worn by the spreader. As the reduction is expressed in terms of logarithmic viral load (tested both as PFU/mL and RNA viral copies/mL), it can be easily observed that the percentage of filtration performance should be high, especially when infectious virion numbers are considered. Indeed, viral RNA can be derived from non-infective viruses or damaged virions. The results reported are quite interesting considering that the presence of the N95 mask only on the receiver slightly reduced the collection of the virus on the gelatin filters placed after the mask and that the greater reduction was determined when the mask was fitted with “adhesive tape” (a condition that does not sum up well a real situation). Nevertheless, the very close space in which the mannequins are placed is a limitation of the study by Ueki et al. (0.24 m^3^) and of our study protocol that should be considered in the evaluation of circulating viruses. In comparison with the Ueki simulator, the proposed aerosol chamber coupled with the BLAM generator can allow the modulation of the viral challenge for endonasal filters; moreover, the SARS-CoV-2 collection in the liquid by swirling sampler improves the capability of correct infectivity assessment, with and without personal protective filter, due to reduced mechanical impact between bioaerosol and collection medium [39].

## 5. Conclusions

A first proposal for a procedure and apparatus is presented for assessing SARS-CoV-2 Viral Filtration Efficiency capable of testing endonasal filter performances.

The apparatus could be further implemented for more sophisticated studies on contaminated aerosols, mimicking in more realistic way human respiration and acting as sneeze or cough (e.g., by chest aerosol emission controller), or studying the evolution in time of bioaerosol within the chamber/apparatus.

Having a testing procedure can sustain application and spread eventually to further research and optimization of this highly promising personal protective equipment that presents potential applications in both adult and younger populations.

## Figures and Tables

**Figure 1 ijerph-19-13790-f001:**
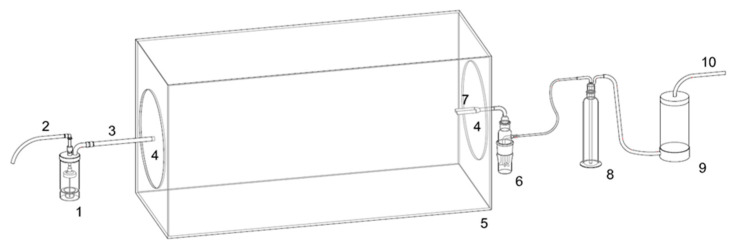
1. CH-Technology BLAM aerosol generator; 2. Connection to compressor (TCR-TECORA Aero Particle Generator) 3. tube connection to 5. Bioaerosol Chamber: 4. side circular bioaerosol chamber closures; 6. SKC BioSampler; 7. Y shape bioaerosol inlet (internal to the bioaerosol box); 8. NaClO trap; 9. Air desiccator/drier; 10. connection to pump (TCR-TECORA BioBravo).

**Figure 2 ijerph-19-13790-f002:**
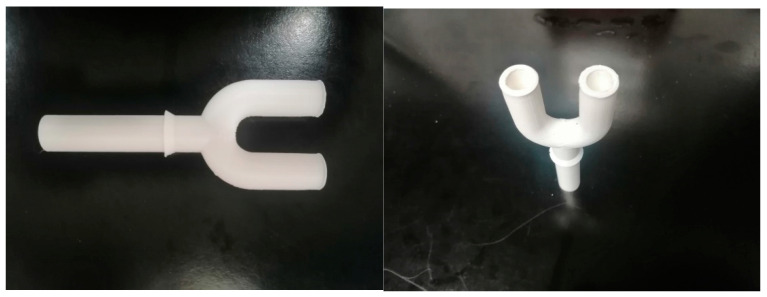
Photographic images of the Y shaped tube sampling port.

**Figure 3 ijerph-19-13790-f003:**
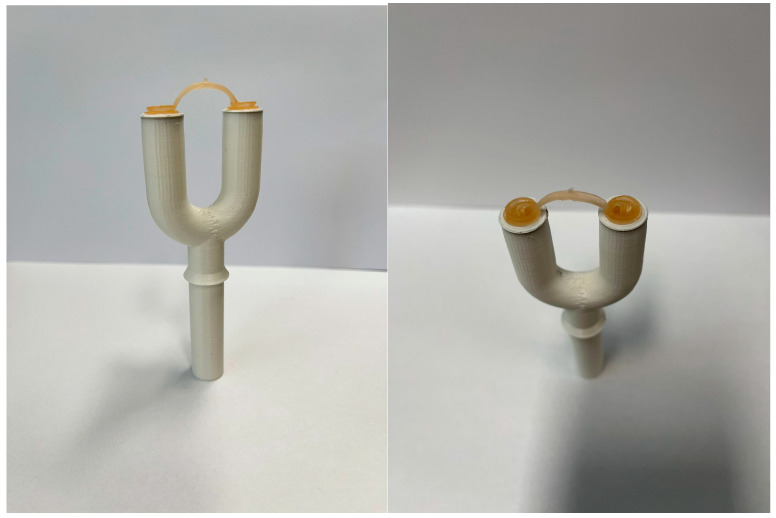
Photographic images of the Y shaped tube sampling port with endonasal filters.

**Table 1 ijerph-19-13790-t001:** Primers and probe for quantification of RNA viral load (qRT-PCR).

Name	Sequence (5′ → 3′)	Concentration	Label
2019-nCoV_N1 Forward primer	GAC CCC AAA ATC AGC GAA AT	500 nM	
2019-nCoV_N1 Reverse primer	TCT GGT TAC TGC CAG TTG AAT CTG	500 nM	
2019-nCoV_N1 Probe	ACC CCG CAT TAC GTT TGG TGG ACC	125 nM	FAMBHQ-1
2019-nCoV_N2- Forward primer	TTA CAA ACA TTG GCC GCA AA	500 nM	
2019-nCoV_N2 Reverse primer	GCG CGA CAT TCC GAA GAA	500 nM	
2019-nCoV_N2 Probe	ACA ATT TGC CCC CAG CGC TTC AG	125 nM	FAMBHQ-1
E gene Forward primer	ACAGGTACGTTAATAGTTAATAGCGT	400 nM	
E gene Reverse primer	ATATTGCAGCAGTACGCACACA	400 nM	
E gene Probe	ACACTAGCCATCCTTACTGCG	200 nM	FAMBHQ-1
Sub-genomic E gene Forward primer	CGATCTCTTGTAGATCTGTTCTC	400 nM

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
