# Peer review of "Operative Protocol for Testing the Efficacy of Nasal Filters in Preventing Airborne Transmission of SARS-CoV-2"

_ijerph, 2022, doi:10.3390/ijerph192113790_

Round 1
Reviewer 1 Report
Many methods have been developed to
sample/analyse airborne viruses - with the
realisation that SARS-COV-2 is airborne.
This is another innovative proposal - though
it is only an experimental protocol so far.
Like other manikin studies, the authors can
consider incorporating the sampling device
(nasal filters) into a human-like head, to
allow the addition of masks and other PPE
devices to be tested see:
https://www.ncbi.nlm.nih.gov/pmc/articles/PMC7227527/
Also, nasal filters may become quickly
blocked by nasal discharges, mucous, etc.
So unless the authors can address this
practical issue, this may be no more than
a lab-based academic exercise.
Author Response
Answer and note to Reviewer
We thank the Reviewer for the helpful suggestions that have allowed us to improve the proposed text.
Reviewer statement 1. "Many methods have been developed to sample/analyse airborne viruses - with the realisation that SARS-COV-2 is airborne.
This is another innovative proposal – though it is only an experimental protocol so far."
Answer 1. We highlight that the proposal – based on our previous publication (doi:10.3390/ijerph182111172) on experimental SARS-CoV-2 aerosolization in a controlled laboratory setting – can address the assessment of viral filtration efficiency by using directly the pathogen of interest and not surrogate species.
Statement 2 of the reviewer. "Like other manikin studies, the authors can consider incorporating the sampling device (nasal filters) into a human-like head, to allow the addition of masks and other PPE devices to be tested see: https://www.ncbi.nlm.nih.gov/pmc/articles/PMC7227527/"
Answer 2. We have modified the text and added the option of using a human-like head for testing masks or other PPEs (see page 3 lines 141-145)
Indeed introducing a human-like head manikin in the aerosol chamber implies cleaning a sampling support with larger surfaces from the nebulized pathogen; this may increase the risks of cross-contaminations among experiments and for operators when handling the used device, and then we consider it is not the elective choice, while aiming at running replicate tests on endonasal filters; we keep the suggestion as option for testing different PPEs.
Statement 3 of the reviewer. "Also, nasal filters may become quickly blocked by nasal discharges, mucous, etc.
So unless the authors can address this practical issue, this may be no more than a lab-based academic exercise."
Answer 3. Main aim of the proposed protocol is proposing a testing system on the viral filtration and inactivation efficacy of endonasal filters, also with a specific virus, and not the usability of PPE. A producer indicates that for endonasal filter products already on the market dealing with allergen filtration, there is indication of usage when nose airways are clean and not on individuals with symptoms of rhinitis (page 3, lines 111-112)
Reviewer 2 Report
line 99: "it is important to consider that younger children show lower tolerance for wearing masks or may not wear them correctly." - Although this is a very interesting work, I don't think a low tolerance for wearing masks should be used to justify the use of nasal filters by children. I really can't see how a more invasive strategy would have a better acceptance, especially in this group of people. line 212-215: "Between the experimental replicates, the BLAM and the SKC biosampler reservoirs are carefully washed with sterile water, moreover, the chamber and the collecting tubes are flushed for 5’ with 75% ethanol and 5’ with sterile water to avoid carryover between the replicates." - Did the authors test for the presence of viral RNA at the end of the SKC Biosampler after each round of replicates? I think this is very important to guarantee that the detection of viral copies was not compromised since ethanol has some known nucleic acid fixative properties. This is specially important during the first set of experiments in which there is no filters in the chamber. Please, state that clear in the text. line 291-292: "Vero E6 cells are cultured in MEM supplemented with 2 mM glutamine, 10% fetal bovine serum, , and 100 U/mL penicillin/streptomycin and seeded on 6 multiwell plates (350.000 cells for well)." - Did the viral infection also occurs in VERO cells with 10% FBS ? High FBS rates such as showed here determine in a competitive inhibition for the viral receptors located at the cell surface, inhibiting viral adsorption in the first place. It is ok to use this percentage to grow the cells, but not during the infection experiments, in which the recommended rates are around 1-2%. Please, make that clear thoughout the text. Did the authors standardize a period of time in which the viral particles stay in contact with the biogel silver nanoparticles in the filters? If yes, please make it clear throughout the text as I wonder that the longer these particles are in contact with the filters, less infectious viral particles are avaiable for the experiments in culture cells.Author Response
We thank the Reviewer for the helpful suggestions that have allowed us to improve the proposed text.
Statement 1 of the Reviewer. line 99: "it is important to consider that younger children show lower tolerance for wearing masks or may not wear them correctly." - Although this is a very interesting work, I don't think a low tolerance for wearing masks should be used to justify the use of nasal filters by children. I really can't see how a more invasive strategy would have a better acceptance, especially in this group of people.
Answer1. The reviewer is right. Younger children may not tolerate endonasal filter.
The sentence was deleted, and we promoted the employment of endonasal filter for adult and adolescent populations.
The text was revised accordingly (page 10, lines 394-400). adding the following paragraph
"Indeed the experimental setting may be modified for assessing filters or masks optimized for children (pediatric applications) and teenagers, by modulating the aspiration of the samplers (less than 12.5 lpm) and using filter holders with a lower section; such variations can modify inner flows in the filters, and the effectiveness of interaction of bioaerosol/airborne pathogens with the AgNPs coated walls. it will be also possible to simulate the inhalation of children, adolescents, adults or elderly, so screening conditions typical of different ages.
Statement 2 of the reviewer. line 212-215: "Between the experimental replicates, the BLAM and the SKC biosampler reservoirs are carefully washed with sterile water, moreover, the chamber and the collecting tubes are flushed for 5’ with 75% ethanol and 5’ with sterile water to avoid carryover between the replicates." Did the authors test for the presence of viral RNA at the end of the SKC Biosampler after each round of replicates? I think this is very important to guarantee that the detection of viral copies was not compromised since ethanol has some known nucleic acid fixative properties. This is specially important during the first set of experiments in which there is no filters in the chamber. Please, state that clear in the text.
Answer 2. - We thank the reviewer for the useful comment.
Yes, we indicate to test presence of viral RNA after each replica (page 6, lines 221, 222). Moreover we have added air flushing after cleaning operation between replicates, in order to remove eventual ethanol residuals from the bioaerosol chamber (page 6, lines 218-219).
Statement 3 of the reviewer. line 291-292: "Vero E6 cells are cultured in MEM supplemented with 2 mM glutamine, 10% fetal bovine serum, , and 100 U/mL penicillin/streptomycin and seeded on 6 multiwell plates (350.000 cells for well)." - Did the viral infection also occurs in VERO cells with 10% FBS ? High FBS rates such as showed here determine in a competitive inhibition for the viral receptors located at the cell surface, inhibiting viral adsorption in the first place. It is ok to use this percentage to grow the cells, but not during the infection experiments, in which the recommended rates are around 1-2%. Please, make that clear thoughout the text.
Answer 3. We thank the reviewer for the observation,
Yes, the infection medium should have 2%FBS.
The text was amended accordingly
Statement 4 of the reviewer. Did the authors standardize a period of time in which the viral particles stay in contact with the biogel silver nanoparticles in the filters? If yes, please make it clear throughout the text as I wonder that the longer these particles are in contact with the filters, less infectious viral particles are avaiable for the experiments in culture cells.
Answer 4 The order of magnitude of the time for air crossing the endonasal filter is estimated to be around 0.001 seconds. Part of the particulate matter carried by the airflow will impact the nano-Ag coated wall, and it will interact with the microbicide biogel (amount of microbes impacting the wall will depend on the diameter of the particles). We expect aerodynamical capture of particles on the filters, with reasonable/good breathability and deactivation of microbial infectivity on endonasal filter surfaces.